# Antiviral Activity of Zinc Oxide Nanoparticles against SARS-CoV-2

**DOI:** 10.3390/ijms24098425

**Published:** 2023-05-08

**Authors:** Stella Wolfgruber, Julia Rieger, Olavo Cardozo, Benjamin Punz, Martin Himly, Andreas Stingl, Patricia M. A. Farias, Peter M. Abuja, Kurt Zatloukal

**Affiliations:** 1Diagnostic and Research Center for Molecular Biomedicine, Institute of Pathology, Medical University of Graz, 8010 Graz, Austria; 2PHORNANO Holding GmbH, Kleinengersdorferstrasse 24, 2100 Korneuburg, Austria; 3Post-Graduate Program on Electrical Engineering, Federal University of Pernambuco, Cidade Universitaria, Recife 50670-901, Brazil; 4Department of Biosciences and Medical Biology, Paris Lodron University of Salzburg, 5020 Salzburg, Austria; 5Department of Biophysics and Radiobiology, Post-Graduate Program on Material Sciences, Federal University of Pernambuco, Cidade Universitaria, Recife 50670-901, Brazil

**Keywords:** SARS-CoV-2, zinc oxide nanoparticles, antiviral

## Abstract

The highly contagious SARS-CoV-2 virus is primarily transmitted through respiratory droplets, aerosols, and contaminated surfaces. In addition to antiviral drugs, the decontamination of surfaces and personal protective equipment (PPE) is crucial to mitigate the spread of infection. Conventional approaches, including ultraviolet radiation, vaporized hydrogen peroxide, heat and liquid chemicals, can damage materials or lack comprehensive, effective disinfection. Consequently, alternative material-compatible and sustainable methods, such as nanomaterial coatings, are needed. Therefore, the antiviral activity of two novel zinc-oxide nanoparticles (ZnO-NP) against SARS-CoV-2 was investigated in vitro. Each nanoparticle was produced by applying highly efficient “green” synthesis techniques, which are free of fossil derivatives and use nitrate, chlorate and sulfonate salts as starting materials and whey as chelating agents. The two “green” nanomaterials differ in size distribution, with ZnO-NP-45 consisting of particles ranging from 30 nm to 60 nm and ZnO-NP-76 from 60 nm to 92 nm. Human lung epithelial cells (Calu-3) were infected with SARS-CoV-2, pre-treated in suspensions with increasing ZnO-NP concentrations up to 20 mg/mL. Both “green” materials were compared to commercially available ZnO-NP as a reference. While all three materials were active against both virus variants at concentrations of 10–20 mg/mL, ZnO-NP-45 was found to be more active than ZnO-NP-76 and the reference material, resulting in the inactivation of the Delta and Omicron SARS-CoV-2 variants by a factor of more than 10^6^. This effect could be due to its greater total reactive surface, as evidenced by transmission electron microscopy and dynamic light scattering. Higher variations in virus inactivation were found for the latter two nanomaterials, ZnO-NP-76 and ZnO-NP-ref, which putatively may be due to secondary infections upon incomplete inactivation inside infected cells caused by insufficient NP loading of the virions. Taken together, inactivation with 20 mg/mL ZnO-NP-45 seems to have the greatest effect on both SARS-CoV-2 variants tested. Prospective ZnO-NP applications include an antiviral coating of filters or PPE to enhance user protection.

## 1. Introduction

In 2019, a new infectious disease, COVID-19, caused by SARS-CoV-2, spread rapidly around the world, posing major challenges for healthcare systems, economy and society. The highly contagious pathogen is mainly transmitted via respiratory droplets and aerosols during direct person-to-person contact [1] and, due to its high stability, via contaminated surfaces [2]. Approaches to combat the pandemic comprise reducing exposure and the use of vaccines, as well as the development and testing of newly developed or already approved drugs that reduce the infectivity of the virus. Mechanisms to interact with the virus include blocking of host cell receptors, inhibiting viral binding to and fusion with cell membranes of host cells, blocking receptor-mediated endocytosis, inhibiting viral enzymes, inhibiting viral nucleic acid synthesis, and inhibiting viral assembly and release [3]. To reduce infections, not only the use of antiviral drugs but also the decontamination of surfaces and personal protective equipment (PPE) is crucial. Surface and PPE disinfection have been reported using ultraviolet radiation, vaporized hydrogen peroxide, heat and liquid chemicals [4]. Most of the described methods damage materials or do not reliably disinfect all parts of the surface, as with ultraviolet radiation that is ineffective on areas shielded from light. Alternatives are the use of more material-friendly methods, such as ozone [5] or nanomaterial coatings.

The field of nanotechnology has grown very fast over the past decade, and nanomaterials are used in a variety of applications, such as electronics, cosmetics, water treatment, textile manufacturing and pharmaceuticals, due to their various physiochemical properties [6,7]. Nanoparticles are characterized by a large surface area compared to their particle size of 1–100 nm [8], which enhances reactivity and interaction with viruses or other agents. They exhibit a broad spectrum of antiviral activities, such as the inhibiting receptor binding of virus particles to host cell surface structures, degrading viral particles through the generation of reactive oxygen species (ROS) or interacting with the viral envelope and degrading its structure [9,10,11]. These properties make them highly interesting candidates for virus neutralization with therapeutics or surface coatings [10].

Antiviral nanomaterials can be classified into the following groups: metal nanomaterials, metal oxide-based nano-photocatalysts and nonmetallic nanomaterials [12]. Metallic nanomaterials like Ag, Cu, Au and Fe exhibit broad-spectrum antiviral activities which were described in different studies [13,14,15,16]. They can inhibit viral entry into the host cells, inactivate the viruses before cellular entry or can enter cells and block viral replication by inactivating viral nucleic acids [11,13,16,17]. 

Metal oxide-based nano-photocatalysts, such as ZnO or SnO_2_, are often semiconductors and have an energetic band gap between the valence band (VB) and the conduction band (CB). When the photocatalyst is illuminated at a suitable wavelength, electrons (e^−^) can be excited from the VB into the CB if the energy is greater than the band gap. Thereby, a positively charged hole (h^+^) in which oxidation can take place is left in the VB of the photocatalyst. Hydroxyl radicals (^•^OH) can be formed by a reaction with water. The electrons in the CB induce reductions of absorbed oxygen atoms. During the photocatalytic process, superoxide radicals (^•^O_2_^−^) and other reactive oxygen species (ROS) are formed [18]. The metal oxide-based nanostructures can thus cause physical damage to the virus structure [12,19,20]. The antiviral activity of materials such as titanium dioxide (TiO_2_), tungsten trioxide (WO_3_), copper (II) oxide (CuO), zinc oxide (ZnO) and tin (IV) oxide (SnO_2_) against enveloped and non-enveloped viruses, including SARS-CoV-2, has been described in previous studies [21].

The advantage of nonmetallic nanomaterials is their low toxicity compared to most metallic materials, which are heavy metals that may be harmful to human health and the environment. Their antiviral properties are based on electrostatic adsorption, nanometer size effect [22] and photocatalytic oxidation, similar to metal nanoparticles. Materials used include carbon nanotubes, graphene-based materials and graphitic carbon nitrides [12]. ZnO is generally recognized as safe by the FDA [23] and was described as a promising versatile inorganic material with a broad range of applications [8]. The antiviral activity of ZnO was already described for many different viruses, such as rhinovirus [24], human immunodeficiency virus (HIV) [25], hepatitis E and hepatitis C [26], influenza [27], herpes viruses [28] and, recently, SARS-CoV-2 [29]. In addition to their antiviral activities, ZnO nanoparticles (ZnO-NPs) have antimicrobial and UV-blocking properties and are used in many products, such as cosmetics, food, textiles, electronics and also in PPE [30]. 

Compared to current knowledge on the antibacterial activity of ZnO-NPs [8], data on their antiviral activity are limited. Therefore, the aim of the present study was to investigate the antiviral activity of two ZnO-NPs, ZnO-NP-45 and ZnO-NP-76, against SARS-CoV-2 including relevant variants in cell culture-based virus neutralization assays. The nanoparticles tested were prepared using an environmentally friendly, highly efficient method. The synthesis uses salts (nitrates, chlorates and sulfates) as a starting material and whey as a chelating agent instead of conventional analytical reagents [31]. The results of this research may form the basis for the development of an antiviral coating for PPE and filters to improve protection and reduce the transmission of pathogens such as SARS-CoV-2.

## 2. Results

### 2.1. Metabolic Interference Assay

To investigate the metabolic effect of ZnO-NPs on Calu-3 cells, we performed the resazurin reduction assay [32,33,34] to determine nanoparticle concentrations that do not lead to cytotoxic or metabolic effects. This is crucial since too high concentrations could inhibit cell function, which could also interfere with virus replication in the cells and could lead to false positive results in the neutralization assays. The assay was performed under the same experimental conditions as the virus neutralization assays but without SARS-CoV-2. In brief, the metabolic activity of Calu-3 cells was measured 48 h after treatment with suspensions of ZnO-NP-45 and ZnO-NP-76 at seven different concentrations (1 mg/mL–100 mg/mL) and was normalized to the metabolic activity of untreated cells (Figure 1). The workflow is illustrated in Appendix A.

ZnO-NP-45 and ZnO-NP-76 had no major effect on metabolic activity and viability of the cells at concentrations of up to 20 mg/mL. Higher concentrations (40 mg/mL–100 mg/mL) led to reduced metabolic activity of the cells, but even at 100 mg/mL ZnO-NP-45 and ZnO-NP-76, the cells still showed approximately 30–35% of metabolic activity, compared to untreated cells.

Based on these results, 20 mg/mL, 10 mg/mL and 5 mg/mL ZnO-NP-45 and ZnO-NP-76 were selected for the SARS-CoV-2 neutralization assays.

### 2.2. Characterization of Zinc Oxide Nanoparticlets

ZnO-NPs were synthesized as described in Material and Methods and characterized using UV-VIS spectroscopy, scanning transmission electronic microscopy (STEM) and dynamic light scattering (DLS).

UV-VIS spectroscopy is often used to determine the size of nanoparticles. In the case of ZnO as a semiconductor, the initial and final absorbance wavelengths are closely related to the band gap of the particles, which is inversely proportional to their size. UV-VIS spectroscopy (Figure 2) comparing the two nanomaterials presented a faster decay in absorbance with increasing wavelength in the case of ZnO-NP-45, indicating the presence of smaller nanoparticles in ZnO-NP-45, compared to ZnO-NP-76 (Figure 3).

The hydrodynamic sizes and zeta potentials of the nanoparticles were determined using DLS and were measured with the ZetaSizer Nano ZS. Both ZnO-NP-45 and ZnO-NP-76 exhibited a positive zeta potential (28.87 ± 0.45 and 14.10 ± 0.20, respectively). In contrast, the reference particles exhibited a negative zeta potential (−1.69 ± 1.27) (Table 1). Although positively charged nanoparticles have been reported to be more cytotoxic than negatively charged particles [35], we did not observe cytotoxic effects at concentrations up to 20 mg/mL, as confirmed by the metabolic interference assay (Figure 1).

The size distribution of the nanoparticles was determined by DLS weighted for number and intensity (Appendix A). The intensity-weighted values as well as the number-weighted values, indicate the presence of agglomerates in suspension. The high PDI values show the polydispersity of ZnO-NP-45 and ZnO-NP-ref. ZnO-NP-76, in contrast, was monodispersed, as indicated by a PDI of 0.3 (Table 1).

STEM images of ZnO-NP-45 displayed a polydisperse mix with a size distribution of 45 ± 15 nm (Figure 3A,B). Images of the ZnO-NP-76 nanomaterial showed particles of about 76 ± 16 nm (Figure 3C,D). Both ZnO-NPs were observed to form agglomerates in suspension. The reference nanomaterial ZnO-NP-ref were NPs of <100 nm and also tended to form agglomerates in suspension. All nanoparticles are shown with the maximum feasible magnification. The size differences of the nanoparticles between STEM and DLS are a result of their strong agglomeration. Energy dispersive X-ray analysis (EDX) intensity maps for the three NPs are shown in Appendix A. Therefore, based on this analysis, NPs were free from trace metals such as Al, Ag or other heavy metals and offer biocompatibility and low cytotoxicity.

### 2.3. SARS-CoV-2 Neutralization Assays

#### 2.3.1. Antiviral Activity of ZnO-NPs on SARS-CoV-2 Delta

The virus-neutralizing activities of the two different ZnO-NPs were analyzed in three independent assays, summarized in Figure 4a–d. The detailed experimental procedure is illustrated in Appendix A. For the neutralization assays, the NPs were pre-incubated with the virus for 1 h at 37 °C and subsequently centrifuged. Samples were taken from the supernatant of each tube to determine the virus input (VI) used for the infection of Calu-3 cells. The amount of virus RNA at 48 h after infection was determined via RT-qPCR in the cell culture supernatant, and virus copy numbers were calculated using an international reference standard. In addition, a commercial ZnO nanomaterial (Sigma-Aldrich; ZnO-NP-ref) was analyzed as a reference sample at concentrations of 20 mg/mL, 10 mg/mL and 5 mg/mL, respectively (Figure 4e,f). As a positive control for all assays, cells were infected with the same virus copy numbers as pre-incubated with the NPs but without NP pre-incubation. Non-infected cells were used as negative control.

All virus copy numbers are available in the Appendix A.

##### ZnO-NP-45

The pre-incubation of SARS-CoV-2 Delta with different concentrations of ZnO-NP-45 resulted in a concentration-dependent reduction in the number of viral copies in the supernatant after centrifugation (VI) as compared to the copy numbers in the untreated positive control (pos Ctrl) (Figure 4a). The dose-dependent reduction of the virus particles may be explained by the adsorption of the viruses to the NP surfaces and the loss of copies after centrifugation due to precipitation. Virus particles present in the supernatant were further tested for their infectivity by using the supernatant as virus input (VI) for infecting Calu-3 cells. At 48 h after infection, the virus concentration was measured by RT-qPCR in the cell culture supernatant to assess virus replication (Figure 4b). Virus concentrations in the cell culture supernatant of infected cells were reduced by more than 10^6^ times after treatment with 20 mg/mL ZnO-NP-45 in all samples compared to cells infected with the virus without NP-pre-treatment (pos Ctrl). Treatment with 10 mg/mL ZnO-NP-45 also led to a reduction of replicating virus numbers, but not in all tested samples. At a concentration of 5 mg/mL ZnO-NP-45, a larger variation was observed in the individual experimental replicates, ranging from strong virus inactivation to high replication comparable to the positive control.

##### ZnO-NP-76

As already detected with ZnO-NP-45, pre-incubation of SARS-CoV-2 Delta with ZnO-NP-76 resulted in a similar dose-dependent reduction of virus copies in the VI after centrifugation, compared to the not pre-treated positive controls (Figure 4c). The effect seemed to be concentration-dependent since it was observed for all three assays and both NP preparations. Although the adsorption of the viral particles to ZnO-NP-45 and ZnO-NP-76 was comparable, they showed different neutralizing effects. Pre-treatment of SARS-CoV-2 Delta with 20 mg/mL ZnO-NP-76 showed similar virus-neutralizing activity to 10 mg/mL ZnO-NP-45, with a reduction of virus copies in many, but not all experimental replicates (Figure 4d).

##### ZnO-NP-ref

To compare the virus-neutralizing activities of the two ZnO-NPs to a commercially available ZnO-NP (ZnO-NP-ref), an additional assay was performed following the same experimental protocol with five replicates for each NP concentration. The reduction of virus particles in the VI can be compared with the reduction after incubation and centrifugation with ZnO-NP-45 and ZnO-NP-76 (Figure 4e). The VI in the positive control (VI pos Ctrl) was lower than in the other assays with ZnO-NP-45 and ZnO-NP-76 for technical reasons. The neutralizing effect of ZnO-NP-ref at 20 mg/mL and 10 mg/mL was comparable to ZnO-NP-76, with a reduction in virus copies in some, but not all, replicates (Figure 4f). No neutralizing effect was observed at 5 mg/mL. High virus copy numbers were detected in the cells infected with the virus without NP pre-treatment (positive control).

#### 2.3.2. Antiviral Activity of ZnO-NPs on SARS-CoV-2 Omicron

To support the results for the SARS-CoV-2 Delta variant, another neutralization assay was performed using SARS-CoV-2 Omicron B.1.1.529. The experimental procedure, the nanomaterials (ZnO-NP45, ZnO-NP-76, ZnO-NP-ref) and the controls (infected cells as positive control and non-infected cells as negative control) were the same as already used for the Delta variant. The results are shown in Figure 5. All virus copy numbers are shown in Appendix A. Two independent methods were used to evaluate the virus-neutralizing activity of the NP. First, viral copies were determined by RT-qPCR in the supernatants of the cells 48 h after infection; then, the cells were immunohistochemically stained with an antibody to the SARS-CoV-2 nucleocapsid protein. In all wells used as a positive control, infected cells were visible in red, as shown in Figure 6. No infected cells were seen in the negative control wells (Figure 6).

##### ZnO-NP-45

As already observed, pre-incubation of SARS-CoV-2 Omicron B.1.1.529 and the nanoparticles resulted in a concentration-dependent reduction of virus copies in the supernatant, which was further used as VI for the neutralization assay (Figure 5a). The replication of SARS-CoV-2 Omicron B.1.1.529 incubated with 20 mg/mL and 10 mg/mL ZnO-NP-45 was finally reduced by a factor 10^6^ after 48 h of cell culture in all samples, compared to the positive control (infected cells without nanoparticle pre-treatment) (Figure 5b). At a concentration of 5 mg/mL ZnO-NP-45, inactivation of the virus was observed in some but not all replicates. The effect ranged from strong virus reduction to high replication, as already detected for SARS-CoV-2 Delta (Figure 4).

##### ZnO-NP-76

Pre-incubation of the virus with ZnO-NP-76 (Figure 5c) reduced virus copies in the VI comparable to the viral adsorption to ZnO-NP-45 (Figure 4a). Pre-treatment with ZnO-NP-76 (Figure 5d) resulted in similar but less constant virus inactivation to that observed with ZnO-NP-45 (Figure 4b).

##### ZnO-NP-ref

As a further control, an additional experimental series was performed with the reference nanoparticles, which led to an inactivation of SARS-CoV2 Omicron in all tested samples in this assay (Figure 5f). The virus particle reduction in the VI is comparable to the reduction after incubation with ZnO-NP-45 and ZnO-NP-76 (Figure 5e).

### 2.4. Immunohistochemistry

To demonstrate the virus-neutralizing effect of the NPs with a second independent read-out in addition to virus copy number calculations based on RT-qPCR, we immunohistochemically (IHC) stained the infected cells with an antibody against the SARS-CoV-2 nucleocapsid (Figure 6). As expected, high numbers of infected cells were visible in all positive control (pos Ctrl) wells. Pre-treatment with ZnO-NP-45 at 20 and 10 mg/mL showed no infected cells, which is in line with the RT-qPCR results. In samples incubated with 5 mg/mL ZnO-NP-45, some wells showed infected cells. This outcome is in line with the virus copy numbers after the RT-qPCR (Figure 5b). For ZnO-NP-76 at 20 and 5 mg/mL, the IHC results confirmed data obtained by RT-qPCR showing both infected and non-infected cells in the stained wells. After treatment with 10 mg/mL ZnO-NP-76, no infected cells were detected. Furthermore, for the three concentrations of the reference nanoparticles, no infected cells were seen after the IHC staining which well reflects the very low virus copy numbers after the RT-qPCR (Figure 5f). All positive control wells were infected, while all negative control wells showed no infected cells.

## 3. Discussion

In this study, we examined the antiviral activity of two novel ZnO-NPs (ZnO-NP-45 and ZnO-NP-76) against the Delta and Omicron variants of SARS-CoV-2 in several independent cell-culture-based experimental series.

These nanomaterials are of great interest due to their physical and chemical properties, low cytotoxicity and sustainable manufacturing process, which allows an environmentally friendly “green” synthesis using salts (nitrates, chlorates and sulfates) as a starting material and whey as a chelating agent instead of conventional analytical reagents. Additionally, the manufacturing process allows doping of the ZnO with other antiviral materials, such as Ag or Cu, which could provide additional antiviral activities for future research and applications. In doped NPs, the atoms of the dopant are located inside the ZnO lattice and not on its surface. To our knowledge, such combinations have not yet been tested.

Our results show that ZnO-NP-45 inhibits replication of SARS-CoV-2 by a factor of up to 10^6^ when incubated at a concentration of 20 mg/mL for 1 h prior to cell infection. This high factor of virus inactivation even exceeds the requirements for disinfectants since, according to the Robert Koch Institute (RKI) and the guideline of the German Association for the Control of Virus Diseases (DVV), a reduction of at least 10^4^ is required for virus disinfection [36]. Lower concentrations of ZnO-NP-45 or ZnO-NP-76 also reduced virus replication, but not in all samples tested. In particular, after treatment with ZnO-NP-76, large differences in virus inactivation were observed at all concentrations tested, ranging from a strong reduction of infectious virus to high replication. This heterogeneity could be explained by incomplete virus inactivation after nanoparticle pre-treatment as well as by secondary infections occurring during the virus neutralization assay. This may be the case because within the 48 h of incubation of infected cells, even a few virus-replicating cells produce sufficient virus particles that may lead to secondary infections of most cells resulting in very high virus copy numbers as determined by RT-qPCR and also seen in IHC.

The reference nanoparticles ZnO-NP-ref also reduced the number of infectious virus particles but seemed to exhibit a stronger antiviral effect on the Omicron variant, as shown in Figure 5. Inactivation of infectious virus was observed for all concentrations and all replicates. With SARS-CoV-2 Delta, only a small number of replicates was inactivated at all NP concentrations. NP-ZnO-ref and NP-ZnO-76 are comparable in their inconsistent antiviral effect on the Delta variant, while 20 mg/mL ZnO-NP-45 resulted in an inactivation of infectious particles by a factor of 10^6^ in all replicates. For the Omicron variant, a constant inactivation using ZnO-NP-45 was observed at 20 mg/mL and also at 10 mg/mL. Therefore, inactivation with 20 mg/mL ZnO-NP-45 seems to have the greatest effect on both SARS-CoV-2 variants tested.

We also noticed poor dispersion of the nanoparticles in the medium, leading to unstable suspensions, which made it difficult to standardize the experimental conditions, which may also contribute to some variations. To reduce this problem, each nanoparticle suspension was mixed immediately before use for each working step. During pre-treatment, the samples were shaken at 300 min^−1^ to keep as many particles as possible suspended and to minimize precipitation before centrifugation.

Pre-treatment of the virus with nanoparticles led to binding and co-precipitation of the virus and nanoparticles after centrifugation. This interaction was dose-dependent and reduced the VI used for the infection assay performed with the supernatant after pre-treatment and centrifugation. Although the pre-treatment and centrifugation reduced virus copy numbers by a factor of up to 10^4^, the reduction of virus replication was up to a factor of 10^6^, which indicates that in addition to adsorption to its surface, the nanoparticles tested reduced virus infectivity by further inactivation. This appears to be a chemical antiviral effect on the virus that could inhibit infectivity when incubated together for one hour. The reference NPs (ZnO-NP-ref) also demonstrated an antiviral effect, as the reduction in viral replication cannot be explained only by particle loss in the VI after pre-incubation and centrifugation. Whether nanoparticles that were not precipitated by centrifugation exert some additional effect on the cells used for the neutralization assay cannot be excluded. However, the metabolic interference assay performed excluded major interference with cell function at concentrations of up to 20 mg/mL nanoparticles.

The different antiviral effects of the two nanoparticle preparations might be explained by their differences in size distribution. As revealed by TEM and DLS, ZnO-NP-45 is composed of particles with sizes from approximately 20 nm to 80 nm, whereas ZnO-NP-76 is monodispersed and contains particles of about 80 nm. The enhanced virus binding and neutralization activity of ZnO-NP-45 might, therefore, be a consequence of the greater total reactive surface than that of ZnO-NP-76.

Previous studies on ZnO-NP have primarily focused on their inhibitory effect on various bacteria. For this reason, data on the effect of ZnO-NP on viruses are less common in the literature and are not yet available for SARS-CoV-2 variants. One possible antiviral effect could be hydrolyzation and inactivation of SARS-CoV-2 bound to ZnO-NPs by the alkaline zinc oxide. Another observation was made for the influenza virus H1N1. ZnO-NPs, both uncoated and PEGylated, were found to inhibit the intracellular replication of the H1N1 virus after cell infection by a mechanism possibly related to the release of Zn^2+^ ions [27]. No comparable observations were made for SARS-CoV-2, but a similar mechanism can be expected and needs further research. The antiviral activity of ZnO-NPs on SARS-CoV-2 was already shown with VeroE6 cells, reporting severe damage to the viral envelope by free radicals causing oxidative stress to SARS-CoV-2 [29]. Reactive oxygen species (ROS) generated through photocatalysis can destroy proteins, lipids, carbohydrates and nucleic acids, ultimately leading to viral inactivation [37].

All of the mechanisms described here could be responsible for inactivating the virus, but there is no specific evidence of which chemical mechanism was responsible for the virus inactivation observed in our experiments.

In conclusion, the findings of this study clearly demonstrate a high antiviral activity of ZnO nanoparticles against the two SARS-CoV-2 variants Delta and Omicron, which is based on adsorption and an additional not yet defined antiviral effect. Particularly the ZnO-NP-45 nanomaterial, which is more active than the ZnO-NP-76, could be a promising material for future usage as a surface coating for antiviral PPE, especially coverall suits, face masks, and antiviral filters that could be used for air-conditioning and room ventilation systems.

## 4. Materials and Methods

### 4.1. Cell Culture

Calu-3 cells (Biomedica, Vienna, Austria) were cultured in Minimal Essential Medium (MEM) (Thermo Fisher Scientific, Waltham, MA, USA) containing 10% fetal calf serum (FCS) (Thermo Fisher Scientific, Waltham, MA, USA), 2% L-glutamine (Merck KGaA, Darmstadt, Germany) and 1% Penicillin–Streptomycin (PenStrep) (Thermo Fisher Scientific, Waltham, MA, USA). VeroE6 cells (Biomedica, Vienna, Austria) were cultured in Minimal Essential Medium (MEM) (Thermo Fisher Scientific, Waltham, MA, USA) containing 5% FCS (Thermo Fisher Scientific, Waltham, MA, USA), 2% L-glutamine (Merck KGaA, Darmstadt, Germany) and 1% PenStrep (Thermo Fisher Scientific, Waltham, MA, USA). All cells were cultured at 37 °C and 5% CO_2_.

### 4.2. Synthesis and Characterization of Zinc Oxide Nanoparticles

The two zinc oxide nanoparticle types, ZnO-NP-45 and ZnO-NP-76, were synthesized using a modified green sol-gel method, as described by Soares et al. in 2020 [31]. In brief, ZnO nanoparticles (NPs) were successfully synthesized by a whey-assisted sol-gel method. The composition of the gel and the subsequent calcination temperature showed efficiency in controlling the growth of nanocrystals. The sustainable method proved to be highly efficient for the synthesis of crystalline ZnO-NPs. As reference, commercially available ZnO-NPs <100 nm particle size (Cat# 544906, Sigma-Aldrich, Saint Louis, MO, USA) were used in separate neutralization assays.

The hydrodynamic sizes and zeta potentials of the ZnO-NPs were evaluated using dynamic light scattering (DLS). A concentration of 100 µg/mL of NPs was prepared in Millipore water, and the measurements were performed using the ZetaSizerNano ZS (Malvern Panalytical, Malvern, UK) and the ZetaSizer Software (Malvern, 7.03, Malvern Panalytical, Malvern, UK), modulating the settings for refractive index of the NP composition and dispersant (2.0 for zinc oxide). The primary particle size and elemental composition of the ZnO-NPs were further determined by scanning transmission electron microscopy (STEM) and energy-dispersive X-ray spectroscopy (XPS). For measurement, 2 µL of a 10 µg/mL NP dispersion was dried overnight on a lacy carbon-coated copper TEM grid and imaged using the JEM F200 (JEOL, Freising, Germany) electron microscope in STEM mode operated at 200 kV. Primary particle size was determined by calculating the mean ± SD of minimum 10 particles via image processing with the ImageJ software (NIH, Bethesda, MD, USA) and manual measuring. EDX intensity maps were acquired with a beam current of 0.1 nA and a beam diameter of 0.16 nm.

### 4.3. Dispersion of Zinc Oxide Nanoparticles

NPs were washed with 70% EtOH and air dried. The required amount of each NP preparation was weighed into 5 mL Eppendorf tubes and suspended in MEM medium, initially without FCS, to avoid unspecific adsorption of the NPs to BSA and other proteins present in the serum. The tubes were mixed for 30 s. using a vortex mixer to minimize agglomeration. It was observed that the nanoparticles precipitate within a short time. Changes in pH or suspension in other solvents did not improve the stability of the NP suspensions. The tubes were mixed before each working step to keep the NPs in suspension during the experiments. During pre-incubation of NPs and SARS-CoV-2, tubes were gently shaken at 300 min^−1^.

### 4.4. Metabolic Interference Assay

The possible metabolic interference of ZnO-NP on Calu-3 (human lung epithelial cells) cells was evaluated using the resazurin reduction assay. Resazurin is a non-toxic dye that is reduced in metabolically active cells to fluorescent resorufin. Its concentration is measured fluorometrically at an excitation wavelength of 530 nm and an emission wavelength of 590 nm [32,33,34]. The optimal excitation wavelength for this assay in our experiments was found to be 485 nm. A stock solution of 1 mM resazurin (Sigma Aldrich R7017-1G, Merck KGaA, Darmstadt, Germany) was prepared in phosphate-buffered saline (PBS) (Thermo Fisher Scientific, Waltham, MA, USA), sterile filtered using 0.2 µm syringe filters (Thermo Fisher Scientific, Waltham, MA, USA) and stored at 4 °C protected from light. To determine the possible metabolic interference, the resazurin assay was performed following the same conditions as the neutralization assays but without the virus (Appendix A). Calu-3 cells were seeded into 48-well microtiter plates (Corning Incorporated, Kennebunk, ME, USA) at a density of 30,000 cells per well and incubated at 37 °C and 5% CO_2_ 48 h before the assay. Suspensions of decreasing ZnO-NP concentrations (100, 80, 40, 20, 10, 5 and 1 mg/mL) were prepared in MEM medium containing 2% L-glutamine and 1% Penicillin–Streptomycin. ZnO-NP were incubated in medium for 1 h at 37 °C and 300 min^−1^ and subsequently centrifuged at 13,000× *g* for 5 min. The supernatant was collected and added to Calu-3 cells at 37 °C and 5% CO_2_ for 1 h. Then, the cells were washed with MEM and fresh medium containing 10% FCS, 2% L-glutamine and 1% Penicillin–Streptomycin, was added. Cells without ZnO-NP treatment were used as references. The plate was incubated at 37 °C and 5% CO_2_ for 48 h. To measure the metabolic activity of the cells, resazurin was added to each well at a final concentration of 10 µM. The fluorescent signal was recorded at a wavelength of 485/590 nm over a time period of 2 h at 37 °C using a microplate reader (BioTek Synergy 4, Szabo-Scandic HandelsgmbH, Vienna, Austria). The cell metabolic activity was measured 48 h after ZnO-NP treatment for each concentration. Linear regression equations of the recorded signals were generated, and the resulting slopes were normalized to the reference and plotted in a graph.

### 4.5. Preparation of SARS-CoV-2 Virus Stocks

All work with infectious SARS-CoV-2 was performed under biosafety level (BSL)-3 conditions [38]. The experimental series was performed using a SARS-CoV-2 Delta virus patient isolate (isolated at the Diagnostic and Research Institute of Pathology, Graz, Austria, GISAID accession number EPI_ISL_4847176 delta-like variant) and SARS-CoV-2 Omicron (Strain HCOV-19/Netherlands/NH-EMC-1720/2021, Omicron variant, Lineage B.1.1.529—Calu 3 culture, Erasmus-MC). For propagation of the SARS-CoV-2 viruses, VeroE6 cells were infected with the different isolates and incubated at 37 °C and 5% CO_2_ for 72 h. To release intracellular viral particles from adherent cells in the cell culture flasks, cells were lysed by a freeze and thaw cycle, followed by a centrifugation step (10 min at 3000× *g*) to remove cell debris and sterile filtered with 0.2 µm syringe filters (Thermo Fisher Scientific, Waltham, MA, USA). Virus stocks were stored at −80 °C until use. The titer of the virus stock was determined using a focus-forming assay. Briefly, VeroE6 cells were seeded in 48-well cell culture plates at a density of 30,000 cells per well and infected with 200 µL of the 10-fold serially diluted virus for 1 h at 37 °C and 5% CO_2_. Each dilution was tested in triplicate. After infection, the cells were washed with MEM, and 500 µL of overlay medium consisting of 1.5% carboxymethylcellulose sodium salt (Merck KGaA, Darmstadt, Germany) in MEM containing 2% FCS and 1% PenStrep was added to each well. After 72 h of incubation at 37 °C and 5% CO_2_, the cells were fixed with 4% neutral-buffered formalin (SAV Liquid Production GmbH, Flintsbach am Inn, Germany) for 30 min followed by antibody staining according to the immunohistochemistry protocol below to determine the number of infected cells.

### 4.6. Immunohistochemistry (IHC)

For immunohistochemical staining, SARS-CoV-2 infected and control cells were fixed for at least 30 min with 4% neutral-buffered formalin (SAV Liquid Production GmbH, Flintsbach am Inn, Germany) and washed 3 times with PBS (Gatt-Koller GmbH, Absam, Austria). The cells were then incubated with 0.1% Triton X-100 (Merck KGaA, Darmstadt, Germany) for 10 min, washed 3 times with PBS and incubated for 30 min in 3% H_2_O_2_ (Merck KGaA, Darmstadt, Germany) dissolved in methanol (Merck KGaA, Darmstadt, Germany) and washed again with PBS. The primary antibody, SARS-CoV-2 nucleocapsid antibody (Sino Biological, Bejing, China, Cat# 40143-V08B), diluted at 1:1000, was added to the cells and incubated at RT for 1 h. The cells were washed 3 times with PBS, and the ready-to-use detection system reagent EnVision Detection Systems, Peroxidase/DAB, Rabbit/Mouse (Agilent Dako, Glostrup, Denmark, Cat# K5007) was added for 30 min, followed by 3 washing steps with PBS. DAB + Chromogen X50 (Agilent Dako, Cat# K5007) was applied to each well to visualize bound secondary antibodies. The reaction was stopped by adding PBS. Cells were washed again with PBS to remove reagent, and fresh PBS was added to keep them humid. Images were taken by light microscope (Nikon, Eclipse, TS100; Nikon Europe BV, Amsterdam, The Netherlands) equipped with a JENOPTIK GRYPHAX^®^ camera (Breitschopf, Innsbruck, Austria). SARS-CoV-2 infected cells appear red after antibody staining.

### 4.7. SARS-CoV-2 Neutralization Assays

To determine the neutralizing effect of ZnO-NP on SARS-CoV-2, Calu-3 cells were seeded into 48-well cell culture plates (30,000 cells/well) two days prior to the assay. 

Based on the cell viability measurement, the following ZnO-NP concentrations were chosen: 20, 10 and 5 mg/mL (in MEM, without FCS). SARS-CoV-2 virus at an MOI (multiplicity of infection) of 0.002 was added to the prepared NP solutions and incubated for 1 h at 37 °C and 300 min^−1^. Afterward, the samples were centrifuged for 5 min at 13,000× *g*, and supernatants were transferred to fresh tubes. To determine the virus input (VI) for cell infection, 140 µL of each supernatant was collected for RNA isolation and RT-qPCR (reverse transcription quantitative polymerase chain reaction). Five wells per supernatant tube were infected for 1 h at 37 °C and 5% CO_2_ to test each NP concentration in five replicates. Virus without NP pre-treatment was used as positive control, and non-infected cells served as negative control for the assay.

After infection, the cells were washed with MEM (no FCS), and fresh MEM (10% FCS) was added to the cells. After 48 h incubation at 37 °C and 5% CO_2_, supernatant of each well was collected to determine virus concentrations at t48 (timepoint 48 h after infection) via RNA isolation and RT-qPCR. A schematic workflow of the neutralization assay is shown in Appendix A.

### 4.8. RNA Isolation and Reverse Transcription Quantitative PCR (RT-qPCR)

SARS-CoV-2 viral RNA from cell culture supernatant was extracted using the QIAamp^®^ Viral RNA Mini Kit (Qiagen, GmbH, Hilden, Germany) according to the manufacturer’s recommendations. RNA samples were eluted with 40 µL Milli-Q water and stored at −80 °C. Viral replication was detected via RT-qPCR using a Rotor-Gene Q thermal cycler (Qiagen) and the QuantiTect^®^Probe PCR Kit (Qiagen). Forward primer 2019-nCoV_N2-F (5′-TTA CAA ACA TTG GCC GCA AA-3′), reverse primer 2019-nCoV-N2-R (5′-GCG CGA CAT TCC GAA GAA-3′) and the N2 probe 2019-nCoV_N2-P (FAM-ACA ATT TGC CCC CAG CGC TTC AG-BHQ1) were obtained from Eurofins Genomics (Ebersberg, Germany). The primer and probe sequences were used as recommended by the Centers for Disease Control and Prevention (CDC) in February 2020 [39]. The total volume of each RT-qPCR reaction was 25 µL. The thermal profile was 50 °C for 30 min, followed by 95 °C for 15 min and 45 cycles of 95 °C for 3 s and 55 °C for 30 s.

### 4.9. Data Analysis

Data analysis, copy number calculations, statistics and graphical presentations were performed with Dotmatics GraphPad Prism 9 (Boston, Massachusetts). Statistical differences between groups were determined using Kruskal–Wallis test corrected for multiple comparisons. Symbol meaning are: ns = *p* > 0.05, * = *p* ≤ 0.05, ** = *p* ≤ 0.01, *** = *p* ≤ 0.001, **** = *p* ≤ 0.0001. The virus copy numbers were calculated using a calibration curve based on a certified RNA standard (VR-1986D^TM^ from 2019 Novel Coronavirus, Lot: 70035624, ATCC, Glasgow, UK). This commercially available standard, containing 4.73 × 10^3^ genome copy numbers per 1 µL, was serially diluted and analyzed by RT-qPCR. The resulting cq-values were plotted against ln[copy numbers], and the equation obtained from a simple linear regression analysis was used to calculate the copy numbers from the cq-values. The standard curve to calculate between cq-values and virus copy numbers is the equation:y = 1.51x + 38.4

## Figures and Tables

**Figure 1 ijms-24-08425-f001:**
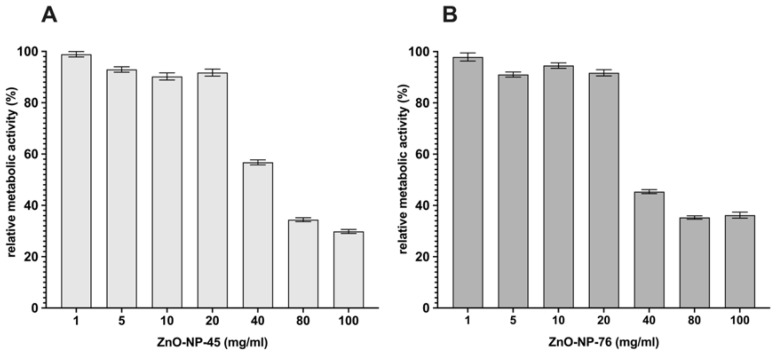
Metabolic Interference Assay of Calu-3 cells. Metabolic activity of Calu-3 cells after treatment with increasing concentrations of ZnO-NP-45 (**A**) and ZnO-NP-76 (**B**). Calu-3 cells were seeded into 48-well plates 48 h before the assay. ZnO-NPs were dispersed in modified Eagle’s medium (MEM) (without fetal calf serum (FCS)) and incubated for 1 h at 37 °C under continuous shaking at 300 min^−1^. Subsequently, the suspensions were centrifuged, and the supernatant was applied to the cells for 1 h at 37 °C and 5% CO_2_. Then the cells were washed, fresh medium (+10% FCS) was added, and the metabolic activity was measured with a resazurin assay after 48 h incubation at 37 °C and 5% CO_2_. Columns represent the relative metabolic activity of the cells (%) normalized to the metabolic activity of control cells without treatment.

**Figure 2 ijms-24-08425-f002:**
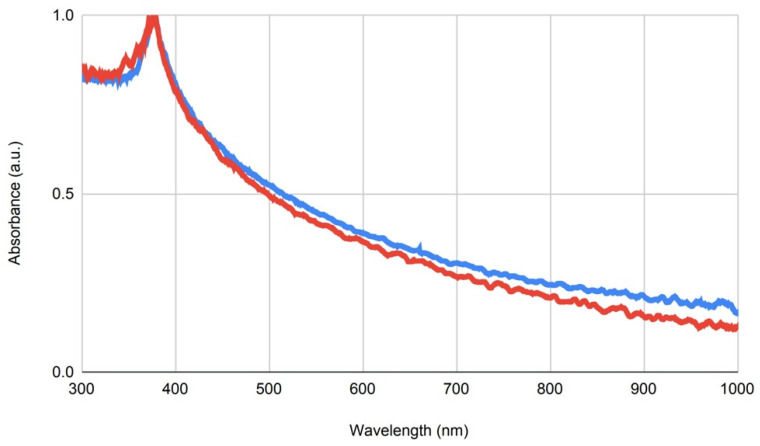
UV-VIS spectroscopy of the two ZnO-NPs. Comparison of ZnO-NP-45 (red) and ZnO-NP-76 (blue), normalized to their respective peak absorbance.

**Figure 3 ijms-24-08425-f003:**
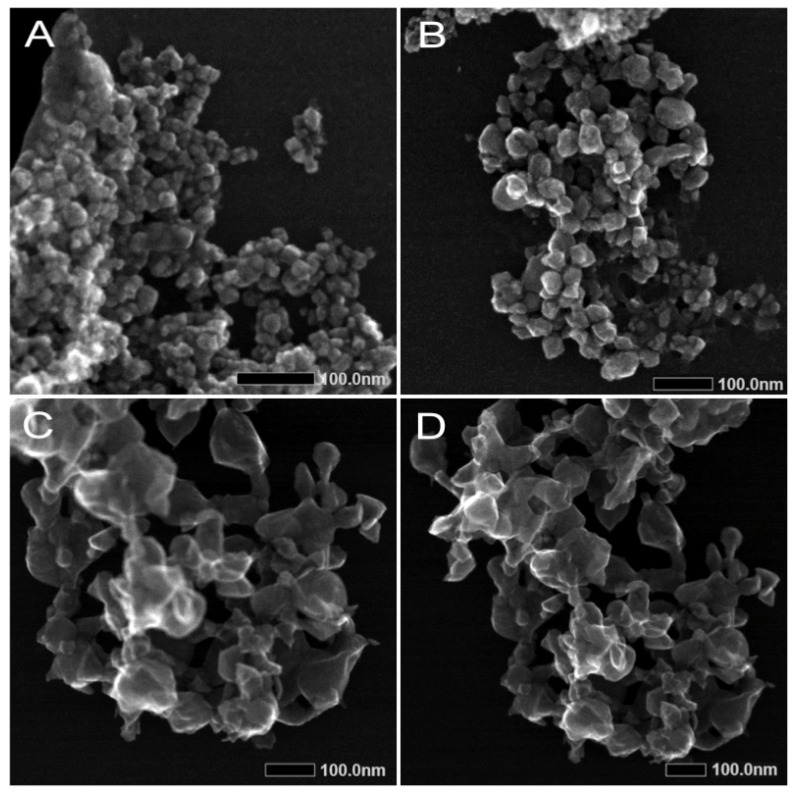
Scanning Transmission Electron Microscopy (STEM) images of ZnO-NP-45 (**A**,**B**) and ZnO-NP-76 (**C**,**D**). The nanoparticles are visible in various sizes and aggregation status. Magnification: (**A**) 400,000×, (**B**) 300,000×, (**C**) 250,000×, (**D**) 200,000×, Size bars: 100 nm.

**Figure 4 ijms-24-08425-f004:**
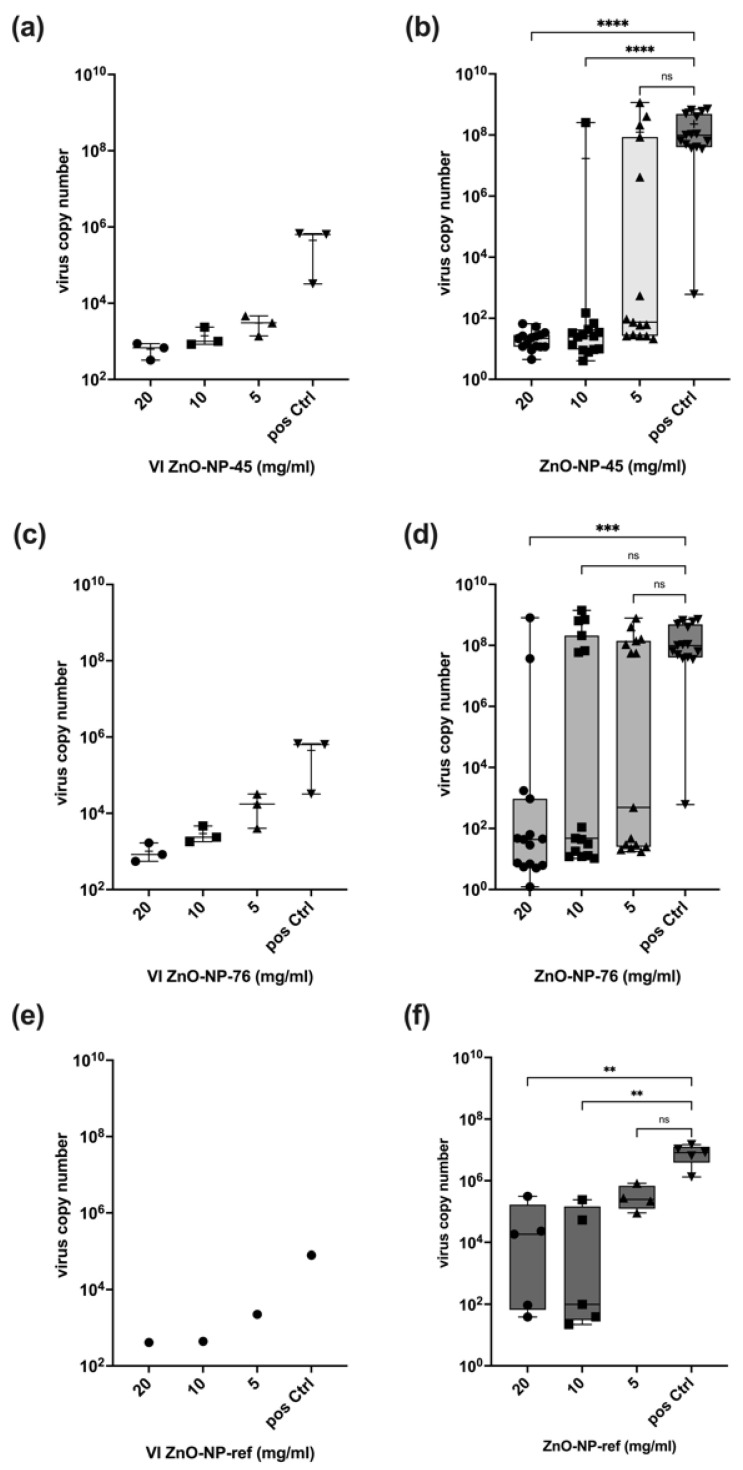
Virus neutralization series with SARS-CoV-2 Delta. (**a**) Summary of virus input (VI) in three independent experimental series used for cell infection after pre-treatment of virus with ZnO-NP-45 for 1 h and subsequent centrifugation. The virus copy numbers in the supernatant were reduced compared to the positive control. The effect was concentration-dependent. (**b**) Summary of virus copy numbers at 48 h post-infection of cell culture with the virus-containing supernatant (VI) after pre-treatment with ZnO-NP-45. ZnO-NP-45 at a concentration of 20 mg/mL led to >10^6^-fold virus inactivation in all tested samples (three experimental series and 5 replicates each) compared to untreated cells (pos Ctrl). ZnO-NP-45 concentrations of 10 and 5 mg/mL also reduced the amount of infectious virus, resulting in variations in the virus concentrations in the experimental replicates. (**c**) Summary of VI in three experimental series used for cell infection after pre-incubation with ZnO-NP-76 for 1 h and subsequent centrifugation. The virus copy numbers in the supernatant were reduced compared to the positive control. The effect was concentration-dependent. (**d**) Summary of virus copy numbers at 48 h post-infection of cell culture with the virus-containing supernatant after pre-treatment with ZnO-NP-76. (**e**) Virus input used for cell infection after pre-incubation with the reference material ZnO-NP-ref. The virus copy numbers were reduced compared to the positive controls as observed for ZnO-NP-45 and ZnO-NP-76. The effect was concentration dependent. (**f**) Virus copy numbers at 48 h post-infection with virus suspension after pre-treatment with ZnO-NP-ref. Infectious virus copies were reduced with 20 mg/mL and 10 mg/mL ZnO-NP-ref compared to untreated cells (pos Ctrl) but not in all replicates. ZnO-NP-ref at 5 mg/mL had no inactivating effect on the virus. *p*-values: > 0.05 = ns, ≤ 0.01 = **, ≤ 0.001 = ***, ≤ 0.0001 = **** (Kruskal-Wallis test).

**Figure 5 ijms-24-08425-f005:**
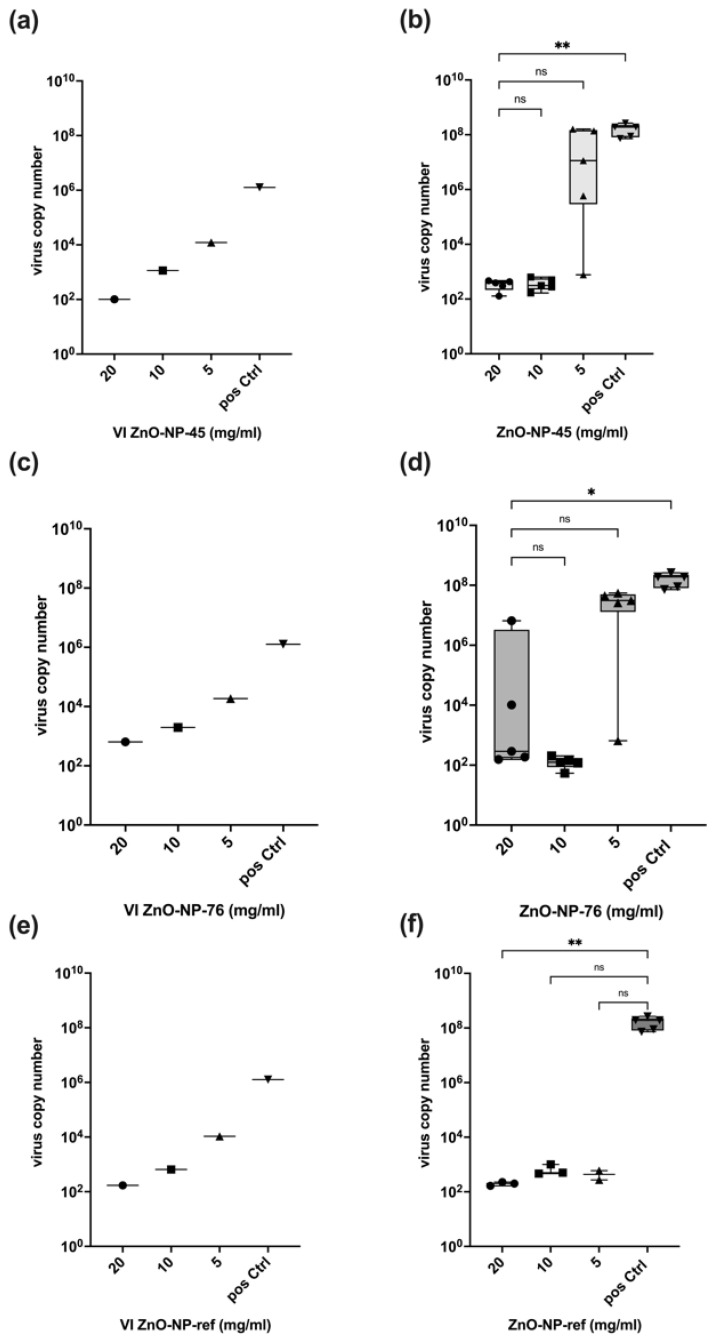
Results of virus neutralization assay with SARS-CoV-2 Omicron after pre-treatment with ZnO-NPs. (**a**,**c**) Virus Input (VI) used for cell infection after pre-incubation with ZnO-NP-45 and ZnO-NP-76 for 1 h and subsequent centrifugation. Virus copy numbers in the supernatant (VI) were reduced compared to the positive control. The effect was concentration-dependent. (**b**) Virus copy numbers after pre-treatment with ZnO-NP-45 at three different concentrations at 48 h post-infection of cell culture with the virus-containing supernatant (VI). ZnO-NP-45 at concentrations of 20 mg/mL and 10 mg/mL led to a virus inactivation by a factor of more than 10^6^ in all tested samples compared to untreated cells (pos Ctrl). At 5 mg/mL, the amount of infectious virus was also reduced, but not for all tested samples. (**d**) ZnO-NP-76 at 10 mg/mL reduced virus replication after pre-treatment by a factor of 10^6^ in all samples. At 20 mg/mL and 5 mg/mL, some, but not all, replicates were inactivated. (**e**) VI used for cell infection after pre-treatment with ZnO-NP-ref for 1 h and centrifugation. The virus copy numbers were reduced compared to the positive control due to adsorption and precipitation. The effect was concentration-dependent. (**f**) Virus copy numbers after treatment with ZnO-NP-ref in three different concentrations at 48 h post-infection of cell culture with the virus-containing supernatant. All concentrations tested led to a reduction of virus replication by a factor of approx. 10^6^. *p*-values: > 0.05 = ns, ≤ 0.05 = *, ≤ 0.01 = ** (Kruskal-Wallis test).

**Figure 6 ijms-24-08425-f006:**
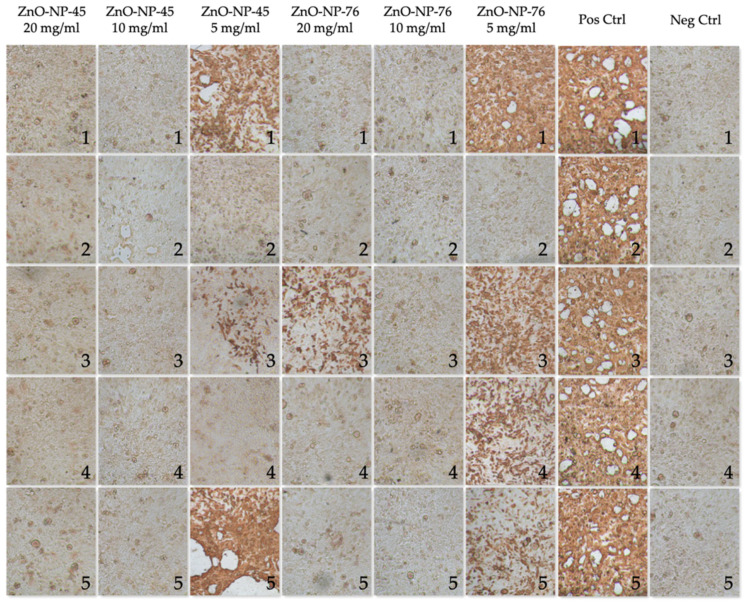
Immunohistochemical staining of Calu-3 cells after virus infection with and without (positive control, Pos Ctrl) ZnO-NP pre-treatment. Calu-3 cells at 48 h after infection with ZnO-NP-45 and ZnO-NP-76 pre-treated virus samples in different concentrations after staining with an antibody specific to the SARS-CoV-2 nucleocapsid. Not infected cells served as negative control (Neg Ctrl). Numbers indicate the wells of the individual replicates per condition; infected cells appear red after the staining. Magnification: 40×.

**Table 1 ijms-24-08425-t001:** Hydrodynamic sizes and zeta potential of ZnO-NP-45, ZnO-NP-76 and ZnO-NP-ref. DLS: Average ± SD of mean values from intensity- and number-weighted distribution analyses of three measurements. Zeta potential: average ± SD in mV determined when suspended in Millipore water. PDI: Polydispersity index indicating the size homogeneity ranging from 0 (all particles are the same size) to 1 (highly polydisperse sample with wide range of particle sizes).

Sample	Size (nm) DLS Intensity	Size (nm) DLS Number	Zeta Potential (mV)	PDI
ZnO-NP-45	596.33 ± 29.10	193.17 ± 9.95	28.87 ± 0.45	0.65
ZnO-NP-76	2125.33 ± 196.59	1451.33 ± 20.17	14.10 ± 0.20	0.30
ZnO-NP-ref	678.27 ± 178.51	230.83 ± 42.25	−1.69 ± 1.27	0.76

## Data Availability

The data presented in this study are available on request from the corresponding author.

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
