# Peer review of "Antiviral Activity of Zinc Oxide Nanoparticles against SARS-CoV-2"

_ijms, 2023, doi:10.3390/ijms24098425_

Round 1

Reviewer 1 Report

Summary: This manuscript evaluates the antiviral functionality of ZnO-NPs against two variants of SARS-CoV-2 (namely Delta and Omicron). The need to expand the search for therapies against emerging pathogens (bacterial, viral, or fungal) beyond small molecular compounds is sufficiently high to justify the work presented.  The following major and minor issues should be addressed before consideration of publication.    

Major issues:

There are numerous preparations of ZnO-NPs that include sol-gel and green synthesis.  While the exact synthesis procedure described herein may be unique, it is not clear that these NPs are truly novel as stated by the authors. This point is made moot by the fact that other ZnO-NPs have shown inactivation activity against SARS-CoV-2 and the fact that a commercially available ZnO-NP from Sigma had the most consistent antiviral activity. While it important to demonstrate reproducibility of anti-SARS-CoV-2 activity of ZnO-NPs in general, to build support to advance toward application, the novelty/impact of this work not very high. This reviewer will leave it to the editorial office to decide on the magnitude of the impact of the work.

The second paragraph of the discussion attributes the differential effects of the two ZnO-NP suspensions to difference in size. However, given the polydispersity of each preparation and high agglomeration potential this is a problematic argument.  This is further refuted by a commercially available NP preparation (<100nm) that was the most consistently effective.  The authors should provide additional characterization data of the ref ZnO-NP to better understand the role of polydispersity vs agglomeration as the role fo the differences in NP effect.

Specific issues:

There was an issue with the referencing of figures that make it difficult to link portions of the text with the specific figure.  There were a couple of instances where the reference was, “Error! Reference source not found”.  These should be corrected before publication

In the discussion of potential mechanisms of action there a multiple instances of reference to viral membrane.  This is an incorrect terminology.  Virus can be enveloped (SARS-CoV-2) which is a portion of the host cell membrane but is still referred to as the envelop not the membrane.  This should be differentiated from the viral capsid.  Please be caution in the use of correct terminology regarding the target viral structure. 

Reviewer 2 Report

This an interesting study, the paper is generally well written and structured. however in my opinion this research paper needed to improve the following:

1. Abstract: is lacking to the objective, therefore a little background is required. Overall, the abstract should be expanded.

2. The introduction is devoid of the aim.

3. in figure 2 the image's magnification is 200 nm and the resoluation is clear.  Figure 3's magnification is different from Figure 2's, and the resolution needs to be increased.

4. Please use the same format for referencing the figures in the body of the manuscript. For instance, you wrote (Figure 7,A) in one paragraph and (Figure 7,B) in another. hence, confirm with the journal instructors.

5. For the characterization of nanoparticles, the FTIR test is essential. you should do it.

6. Nanoparticle concentrations should be expressed in mg/ml rather than as a percentage.

Reviewer 3 Report

The purpose of the manuscript was the analyzis of the antiviral activity of two novel zinc-oxide nanoparticles (ZnO-NP) against SARS-CoV-2 variants Delta and Omicron. The authors tested two nanomaterials differ in size, with ZnO-NP-45 representing particles from 30 nm to 60 nm, and ZnO-NP-76 representing particles ranging from 60 nm to 92 nm. The article is an interesting piece of work and may be interesting to the broader scientific community. The experimental analysis was well designed, the content of the state of art provides useful information about the topic. My recommendation is to accept the article for the possible publication in “International Journal of Molecular Sciences” after the minor revision.

Authors need to revise the methodology section.

Lines 139-140

The authors have written: ‘‘The virus neutralizing activities of the two different ZnO-NPs were analyzed in three independent assays, summarized in...“. The authors have to add the source or the description of the method.

Reviewer 4 Report

The authors are to be commended on some fine work.  Still there is much work to be done to improve the manuscript prior to publication.  In general I would like to see the data presented a little more clearly and more compellingly.  The authors should consider including controls in some experiments and whether the experimental design or methods need improvement? For example, whether the cell cytotoxicity experiments use same conditions as the antiviral experiments?  There needs to be another assay or another way to quantify the antiviral effect besides virus copy number.  A general comment is whether the data can be re-formatted/re-plotted in a way that makes it more compelling or shows significant differences?   The two different sizes of ZnO, that's fine, but how about some other non-ZnO controls, or show the plots in such a way as to present convincing and significant results.  The TEM images for both sizes should be at the same scale, one is out of focus, and there should be an independent means of characterization of their size, PDI and zeta potential.  The UV are not compelling, what significance does this have?  Is there any data on the NPs binding to virus or to the cell surface. One other type of different compelling data needs to be shown. The abstract is too general and the results need to be directly written about the data, saving the conclusions and discussions until last.

Round 2

Reviewer 1 Report

Concerns addressed by authors

Author Response

Reply: We would like to thank the reviewer again for the careful and insightful review of our manuscript and the efforts towards improving it. Below, we would like to respond to the comments and the improvements we made to address the remarks.

To further improve the manuscript the abstract has been updated again and in addition, all changes were proofread by a native speaker to improve the quality of the language.

Reviewer 4 Report

The manuscript looks improved in my opinion, the virus titer and immunhistochemistry experiments are well done.  The abstract still does not accurately reflect, summarize and quantify the results.  Abstract needs to be improved prior to publication. 

Author Response

Reply: We would like to thank the reviewer again for the careful and insightful review of our manuscript and the efforts towards improving it. Below, we would like to respond to the comments and the improvements we made to address the remarks.

The abstract has been extended incorporating the requested further details on the study, providing more context on the novelty of the study, quantified measures on the different behaviors of the three tested preparations, commenting the variations found with ZnO-NP-76 and the differences towards the virus variants. These changes have significantly contributed to better present and summarize the results of this study. In addition, the entire manuscript was proofread by a native speaker to improve the quality of the language.